# Ancient Diseases in Vertebrates: Tumours through the Ages

**DOI:** 10.3390/ani14101474

**Published:** 2024-05-15

**Authors:** Andreia Garcês, Isabel Pires, Sara Garcês

**Affiliations:** 1Exotic and Wildlife Service, Veterinary Hospital University of Trás-os-Montes and Alto Douro, Quinta dos Prados, 4500-801 Vila Real, Portugal; 2CECAV, Centre for Animal Sciences and Veterinary Studies, Associate Laboratory for Animal and Veterinary Science—AL4AnimalS, University of Trás-os-Montes e Alto Douro, 5000-801 Vila Real, Portugal; ipires@utad.pt; 3Earth and Memory Institute, 6120-750 Mação, Portugal; saragarces.rockart@gmail.com; 4Polytechnic Institute of Tomar (IPT), Geosciences Center (UID73), 2300-000 Tomar, Portugal; 5Geosciences Centre, University of Coimbra (u. ID73–FCT), 3001-401 Coimbra, Portugal

**Keywords:** fossil, paleo-oncology, tumour, disease, palaeontology

## Abstract

**Simple Summary:**

Simple Summary: Our research, made possible by recent advancements, has led to more accurate diagnoses of ancient pathologies, despite the rarity of well-preserved specimens, the predominance of bone remains, and the difficulty in distinguishing neoplastic from non-neoplastic lesions in fossils. This study compiles reports of tumours in fossilised animals, highlighting that neoplasms are present in various vertebrates and drawing comparisons to modern instances of similar diseases, thereby providing unique insights into the presence of tumours in ancient animals.

**Abstract:**

Paleo-oncology studies neoplastic diseases in fossilised animals, including human remains. Recent advancements have enabled more accurate diagnoses of ancient pathologies despite the inherent challenges in identifying tumours in fossils—such as the rarity of well-preserved specimens, the predominance of bone remains, and the difficulty in distinguishing neoplastic from non-neoplastic lesions. This study compiles reports of tumours in fossilised animals, highlighting that neoplasms are present in a wide range of vertebrates and drawing comparisons to modern instances of similar diseases. The findings underscore the multifactorial aetiology of tumours, which involves genetic, environmental, and lifestyle factors, and suggest that tumours have been around for at least 350 million years.

## 1. Introduction

Palaeopathology is the study of disease and injury in ancient organisms that includes the examination of fossilised tumours [1,2,3]. Paleo-oncology, in particular, is the study of tumoural diseases in the remains of humans and animals [4]. This science can be an essential tool for understanding diet, behaviour, nutritional disorders, locomotor habits, environmental changes, and disease outbreaks, and it contributes to taphonomic and systematic studies [5,6,7,8].

Fossils are preserved remains of a once-living organism (e.g., animal, plant, bacterium or fungus). They can also be traced fossils (ichnofossils) that show evidence of the organism’s behaviour, such as footprints, bite marks or coprolites [9,10]. Usually, the toughest parts of animals and plants become fossils (e.g., bones, shells, exoskeletons). Nevertheless, the environment is occasionally ideal for preserving completely soft-bodied organisms or even an entire ecosystem. There are several methods for fossilisation depending on the region and environmental conditions [11]. Some fossilisation methods include permineralisation and biosimulation, authigenic mineralisation, replacement and recrystallisation, casts and moulds, adpression, and carbonisation. For a specimen to be considered a fossil, usually, it is over 10,000 years old, but this period can be influenced by the environment and the original organic tissue [12].

Paleo-oncology specifically studies neoplastic diseases in the remains of humans and animals [4]. A neoplasm is an abnormal proliferation of cells resulting from errors in cell division regulation [13,14,15]. Its growth is uncoordinated with the normal surrounding tissue and is associated with metaplasia or dysplasia [16]. When this growth forms, a mass is denominated as a tumour [17]. A neoplasm can be benign or malignant and develop in all tissues, animals or botanicals [18,19,20]. The aetiology of tumours is multifactorial, with factors such as environmental stress (toxins, UV radiation), genetics, diet, stress, local trauma, and pathogenic agents (viruses, bacteria) [21,22,23]. Tumours have been described in almost every class of vertebrates, which are more common in domestic animals [16,21,24].

Tumours have been considered rare in fossils [25,26], possibly due to the higher likelihood of predation on sick animals and the nature of the remains preserved and discovered today, predominantly bones [15,26]. This limitation significantly narrows the scope of detectable diseases from fossils, as the array of diseases identifiable through the examination of bone and teeth is minimal compared to the full spectrum of known diseases. Misinterpretation can also arise from normal skeletal variations, leading researchers to incorrect conclusions [1,4]. Also, it is challenging to identify tumours in fossils since non-neoplasia lesions (e.g., cysts, infectious diseases, trauma, rheumatic disease) can produce lesions similar to those created by neoplastic diseases [1]. Normal skeletal variations can lead scientists to error. It is also important to identify whether it is a true pathological alteration (antemortem) or a taphonomic signature (postmortem) [27]. Distinguishing between true pathological alterations that occurred before death (antemortem) and changes resulting from postmortem processes (taphonomic signatures) is crucial. The fossilisation process can alter skeletal remains through chemical (e.g., soil acidity), biological (growth of algae, bacteria, and fungi), and physical agents (mechanical erosion, micro-fractures), as well as damage from postmortem scavengers, all of which can mimic neoplastic disease. Moreover, lesions attributable to cancer may have altered over time, rendering contemporary diagnostic techniques and criteria potentially unsuitable for accurately diagnosing this disease in ancient and fossilised remains [1,26,27].

Fossilised tumours have been identified in various types of animals, including ancient sauropsids, Osteichthyes, and Synapsids [26]. These discoveries contribute to our understanding of the prevalence of diseases in ancient ecosystems, the evolutionary history of diseases, and the potential impact of environmental factors on the health of prehistoric organisms [28,29]. Also, these discoveries bring additional information regarding this pathology in modern animals and can help us better understand their mechanisms [1]. This paper aims to gather and present reported cases of tumours in extinct animals, providing a comprehensive overview that connects the ancient past to the present.

## 2. Types of Tumours in Fossils

Different types of tumours have been described in fossils, most located in bone structures. Animals can develop various types of bone tumours that may originate from different types of bone cells and can be malignant or benign [30]. Table 1 describes the characteristics of the main types of tumours described in this paper for better understanding.

## 3. Technological and Methodological Advances in the Detection of Neoplasia in Fossils

Positive diagnoses of ‘cancer’ in a vernacular sense are almost unknown in the literature [33]. Neoplasms are equally rare in fossil collections [34], but this may reflect the much more significant focus on human tumours than in veterinary medicine [33]. A reliable archaeozoological diagnosis of tumours is one of the most challenging tasks confronting experts [33].

It is essential to understand that cancer diagnosis is complex and that the size of the samples that survived to be examined currently is not representative of the original population. This can lead to errors in estimating cancer populations among prehistoric animal populations [26]. For example, chondrosarcomas (malignant neoplasms of cartilage) are considered rare in domestic animals, and given their association with perishable soft tissue, archaeological manifestations of their primary forms are doubtful. The same happens with tumours originating from the marrow or other soft tissues. Maxillary fibrosarcoma is a particular periosteal form of this cell-level condition, located in most domestic animals in the cranial bones, especially in the maxilla. It may cause the pathological dissolution of bone tissue (osteolysis) and penetration into the bone cortex and may ultimately cause pathogenic trauma and not leave evidence in the bone [33].

In recent years, several technological and methodological advances have greatly enhanced the understanding and identification of these paleo-oncological lesions. Beyond macroscopic osteological diagnostic criteria, new tools such as medical imaging techniques, histological analyses, and biomolecular methods (e.g., aDNA and proteomic studies) have been used to identify these lesions [35]. The main objective is to develop new non-invasive techniques to identify the lesion without damaging the fossil [26,35].

Histology is a valuable technique, not only in soft tissues but also in bone. Studies in dry bone lesions have shown promising results in differentiating between some tumour types. Immunohistochemistry and molecular analysis methods have provided insights into ancient tumours’ cellular and molecular characteristics [36]. Although a powerful tool in paleo-oncology, this technique does have its limitations. One significant challenge is the preservation of ancient tissues over thousands or even millions of years, and the process of fossilisation can obscure cellular structures, making it challenging to identify cancerous cells under the microscope. The scarcity of well-preserved specimens poses a limitation to histopathological studies in paleo-oncology. Fossilised remains with identifiable tumours are relatively rare, and even when such specimens are found, the preservation quality may vary, affecting the accuracy and reliability of histological analysis. Another limitation is the potential for misinterpretation of pathological features in ancient tissues. Without access to fresh tissue samples or comprehensive clinical data, researchers may struggle to distinguish between cancerous lesions and non-neoplastic abnormalities or postmortem changes [37,38].

Advancements in imaging technologies such as computed tomography (CT), magnetic resonance imaging (MRI), and X-ray fluorescence (XRF) have allowed scientists to study ancient remains for signs of cancerous lesions non-invasively. These techniques provide detailed images of bones and soft tissues, helping to identify abnormalities indicative of cancer [39,40,41,42].

Other techniques, including mass spectrometry and stable isotope analysis, have enabled researchers to detect cancer biomarkers in ancient tissues. These techniques can identify specific compounds associated with cancer metabolism or environmental carcinogens, shedding light on the prevalence and causes of cancer in antiquity [1]. Also, advances in DNA sequencing technologies have enabled the extraction and analysis of ancient DNA, preserving DNA in fossilised remains is variable, and DNA damage over time can complicate the genomic analysis of ancient tumours [43].

Next-generation sequencing (NGS) technologies in the future can be an essential tool in paleo-oncology by allowing researchers to extract and analyse ancient DNA from fossilised remains [44]. By sequencing the genomes of ancient individuals, it may be possible to identify genetic mutations associated with cancer and study their evolutionary history [45]. The advance of technology has already allowed the reconstruction of the DNA complete sequence with a 167,770 bp mitochondrial genome from a woolly mammoth (*Mammuths primigenius*) from only 200 g of bone [45]. By analysing the presence and frequency of specific genetic variants associated with cancer in fossilised remains, it will be possible to identify ancient populations that may have been predisposed to certain types of cancer [46]. By studying the evolution of specific gene variants in neoplasia fossils, it is possible to reveal patterns of mutation accumulation over time. Comparing these mutation patterns with those observed in modern cancer genomes can help to identify conserved mutation signatures associated with particular types of cancer [44]. Insights into the molecular mechanisms underlying neoplastic growths in ancient individuals can be obtained by analysing the functional consequences of ancient genetic mutations and their interactions with environmental factors. Understanding the pathways involved in cancer development and progression throughout time is possible. Although no fossil record of Canine Transmissible Venereal tumour (CTVT) in canids has been found in fossils, sequence analysis of the RPPH1 gene and microsatellite analysis indicates that the tumour is more than 6000 years old (around 10,000 years old) in wild wolves and early domesticated dogs. Some authors hypothesise that it originated when dogs were first domesticated through a mutation in the basic genetic material of the histiocytes [44].

## 4. Tumour Descriptions in Fossil Remains

Here, we compile all neoplasms recorded to date in different geological eras. For a better understanding, Figure 1 presents a paleontological timeline from the Proterozoic through the Cenozoic.

### 4.1. Proterozoic (2.5 Billion to 539 Ma) Aeon

The earth was filled with simple eukaryotic organisms during the Proterozoic Aeon (which included the Paleoproterozoic, Mesoproterozoic, and Neoproterozoic Eras). Although an alteration in the DNA and cells was very likely to occur, no fossil registry survived until today regarding any type of pathology in these organisms [47].

### 4.2. Paleozoic (541 to 252 Ma) Era

This is the first era of the Phanerozoic Aeon, the current aeon characterised by the proliferation of complex multicellular life forms. The Paleozoic Era is subdivided into six periods: the Cambrian, Ordovician, Silurian, Devonian, Carboniferous, and Permian periods [48].

About 530 million years ago, during the Cambrian period, the earliest multicellular organisms were primitive algae and invertebrates, eventually followed by arthropods, plants, and Osteichthyes [49]. The Devonian (419–359 million years ago) is known as the “Age of Osteichthyes”, where fishes became highly diverse with early sharks and armoured placoderms [50,51]. Four cases of neoplasia have been described in fossils from this period (Table 1). During the Carboniferous period (358.9–298.9 million years ago), amphibians predominated [52]. The first Synapsid ancestors appeared during the Pennsylvanian sub-period of the late Carboniferous period [40]. Table 2 presents the records of tumours reported to the moment of elaboration of this work from the Paleozoic Era (Figure 2A,B,F).

### 4.3. Mesozoic (252 to 66 Ma) Era

Sauropsids first appeared about 320 million years ago during the Carboniferous period. Their prime was during the Mesozoic Era (252—66 million years ago) [56], often called the “Age of Reptiles”. During this period, dinosaurs and the ancestors of crocodilians and turtles predominated [57]. The Cretaceous–Paleogene extinction event led to the vanishing of non-avian dinosaurs, leaving Sauropsida and modern sauropsids [58].

Table 3 presents the records of tumours reported to the moment of elaboration of this work from this era. Information regarding species, region, type of tumour, and location is provided (Figure 2C–E,G).

### 4.4. Cenozoic (66 MA to 0 MA) Era

After the Cretaceous–Paleogene extinction, synapsid groups diversified into many new forms and ecological niches during the Paleogene (66.0–23.03 million years ago) and Neogene (23.03–2.588 million years ago) [44]. This era is characterised not only by the diversification and dominance of Synapsida but also by Sauropsids, flowering plants, and significant geological and climatic changes [48]. Table 4 describes some of the tumours found in this period, from the Paleocene to the Pleistocene (Figure 2H–L).

The Holocene Epoch is the most recent epoch of the Earth’s geological history, following the Pleistocene Epoch and spanning from approximately 11,700 years ago to the present day. It represents the period since the last significant glacial retreat and the beginning of the current interglacial period [48]. Paleo-oncological records of this period include some domesticated species as follows:

A severely malformed first incisor from a red deer, dated to the 13th–12th millennium BC, was interpreted as a composite odontoma; it was a rare odontogenic pseudo-tumour [105]. An odontoma was observed in a tooth of a Holocene walrus fossil (*Ontocetus emmonsi*) from Alaska (USA) [26].

One case of osteosarcoma in a fossil was recorded in Grotta della Fungaia (Montemaggio, Siena, central Italy). According to the authors [99], the specimen is chronologically framed in the early Neolithic archaeological horizon. It is a distal fragment of a right tibia attributed to *Ovis aries palustris.* It comprises the distal two-thirds of the right tibia of a young individual, with the final 40 mm exhibiting a neoplastic growth characterised by a rough surface rich in osseous spicules and a spongy structure. The proximal margin aligns with a pathological fracture that interrupts the tibial diaphysis and the surrounding neoplastic osseous cuff. Radiographically, the growth occupies the middle third of the tibia and is characterised by diaphyseal osteolysis and a well-developed periosteal cuff of dense new bone that is radially structured. Histologically, the bone displays a polymorphous structure, with new trabecular and woven bone [99]. Additionally, there are alternating extremely dense areas and empty spaces, which, during life, must have been filled by non-mineralized sarcomatous tissue. The histologic analysis also reveals a significant variability in the trabecular dimensions, with the tumoural spongy bone appearing to be constructed from very slender trabeculae and trabeculae of normal size. The specimen’s macroscopic, radiographic, and histologic aspects are typical of osteoblastic osteosarcoma and provide the first clear-cut evidence of osteosarcoma in fossil remains [99].

Another report is of an olecranon process of a horse from Northampton presenting a lytic cavity that was also considered neoplastic [106]. Baker and Brothwell (1980) [106] report one suspect case of a neoplasm on a pig’s ileum [107]. An equine skull of unknown provenance or date with a large osteoma [108] was placed in the College of Dentistry at the University of Iowa at the time of this paper. An osteoma has also been observed in a chicken from Wichen Bonhunt. It is on the anterior portion of the ulna and about the size of a small pea extending outwards into the medullary cavity [108].

Reports from an excavation of Westbridge Friary describe an equine whose bones had many small nodules on their periosteal surfaces [109]. According to the authors, histological examination of three of these lesions showed changes, suggesting that this animal had a multicentric osteoma. Also, a chicken sacrum and pelvis excavated at Lankhills showed multiple spongy outgrowths that highly suggest the bird had myeloma. Rare histological signs of osteosarcoma were diagnosed on the lingual side of the left mandible of a 7.5-year-old Arabian-type stallion recovered from a mid-13th century Cumanian grave in southern Hungary [110]. This tumour, however, was hardly visible macroscopically and could be identified only by the ‘nested’, lacey structure of bone tissue that replaced the supporting strands of trabecular structure in the body of the mandible [109,110].

A primary malignant bone tumour (telangiectatic osteosarcoma) was reported from a canid with a cranial skeletal pathology from an excavation associated with the Przeworsk culture (III c. BC—V c. AD) [111]. A dog skull, an intentional inhumation, was dated to the Roman influence and the Migration period (I—V c. AD) in Lower Silesia, Poland. The dog was a relatively large animal with a shoulder height of approximately 60 cm. Massive bone changes localised on the facial surface of the left maxilla required a multistage diagnostic protocol using traditional macroscopic and morphometric evaluation and modern diagnostic imaging techniques such as digital radiography, computed tomography, and 3D reconstruction [111]. Recently, a case of a dog with osteosarcoma was reported from an early Roman (1st–2nd century A.D.) pet cemetery in Berenike, Egypt [112].

A possible case of maxillary fibrosarcoma in the left viscerocranial region of an early Modern Age skull was reported from Budapest, Hungary. The bone matrix in the facial region was dissolved in a pathological process, resulting in intravitam loss of the right upper left canine tooth [109].

### 4.5. Statistical Analyses

A total of 72 tumours were described in this review. Most tumours were described in sauropsids (*n* = 34, 47%), followed by Synapsids (*n* = 31, 43%). Of the 72 tumours reported, 51% (*n* = 37) were malignant. Osteoma (*n* = 18) was the most common type of tumour observed. In two main classes, Sauropsida and Synapsids, the predominant type of tumour was benign (n = 17) (Figure 3).

## 5. Cancer through Time

Although rare, some cases of tumours in archaeozoological records appear in the literature [26,113]. The oldest benign tumour reported thus far was in an Upper Devonian armoured osteichthyan (Phanerosteon mirabile), an osteoma dating to approximately 300 million years ago [26]. The oldest unequivocal malignant tumour chondrosarcoma was found in an *Allosaurus fragilis*, a Jurassic Period dinosaur from around 155 million years ago [26,99]. The first metastatic cancer was also observed in a Jurassic dinosaur [69]. Unfortunately, the fossil record contains only a limited number of neoplasm cases, with some diagnoses uncertain. Moreover, new diagnostic techniques have reclassified many previously identified cases as non-neoplastic. This scarcity of data from ancient times makes it challenging to fully understand cancer’s historical prevalence and evolution (e.g., [1,11,30]).

Identifying similar tumour types in both fossils and modern animals suggests that certain forms of cancer have deep evolutionary roots. Finding different types of tumours in fossils can provide valuable insights into the health and biology of ancient organisms. The presence of tumours in multiple fossils from a particular species or group of organisms can suggest the prevalence of diseases during that period. It can provide clues about the overall health of populations and the environmental factors that might have influenced disease occurrence. For example, hadrosaurs (duck-billed dinosaurs) seemed particularly affected by tumours [15].

In modern populations of animals and humans, it is shown that neoplasias are related to the genetic characteristics of the host and environmental conditions [114,115]. Environmental conditions and genetics of populations have changed markedly over history, particularly in the last centuries due to human activity. The ecosystem has been severed and damaged due to anthropogenic activity, leading to the increase in some types of tumours such as pulmonary tumours, leading to changes in the epidemiology of cancer [116]. In nonhuman and human fossils and tissues, there seems to be a particular absence of environmentally caused cancer [117]. By studying cancer prevalence and patterns in fossils from different geological periods, it is possible to track changes in cancer incidence over time. This temporal approach allows for examining how environmental factors, such as changes in climate, habitat, pollution levels, or other ecological factors, may have influenced cancer development and progression throughout Earth’s history [1]. In the past, natural carcinogens such as tannins, phenols, and resins could be found in plants eaten as food (e.g., leaves and fruits). These agents could have been responsible for the development of some tumours [11].

Peto’s Paradox refers to the paradoxical fact that, across different species, there is no clear correlation between body size or lifespan and cancer risk [118]. Typically, larger animals have more cells and longer lifespans, which might suggest they would be more prone to cancer because there are more opportunities for mutations to occur. Conversely, smaller animals with fewer cells and shorter lifespans might be expected to have lower cancer rates. However, this is not consistently observed in nature [118,119]. Several hypotheses have been proposed to explain Peto’s Paradox. Larger, long-lived species may have evolved more effective cancer suppression mechanisms to counteract the increased risk associated with their size and lifespan (e.g., elephants have multiple copies of tumour suppressor genes). Larger organisms may have evolved trade-offs that prioritise mechanisms to prevent cancer at the expense of other factors (e.g., elephants have evolved enhanced DNA repair mechanisms to counteract the increased risk of mutations due to their large number of cells). Differences in lifestyle, environmental exposures between species, and cellular biology (metabolic rate, cellular turnover, and telomere length) may also play a role in determining cancer risk across species [1,118].

Peto’s Paradox has implications not only for extant animals but also for extinct species [118,120]. Extinct animals, like their extant counterparts, varied greatly in size, lifespan, and ecological niches [121]. While direct evidence of cancer in extinct animals is rare, paleopathological studies have uncovered fossilised tumours and other pathological conditions suggesting neoplastic growths in ancient organisms. Analysing the frequency and distribution of these pathologies in extinct taxa can offer indirect clues about cancer prevalence and the evolutionary responses to tumour formation over geological timescales [122]. It would be expected that a high prevalence of tumours could be observed in species such as ornithopods, sauropods, ichthyosaurs or plesiosaurs, some of the largest extinct animals on Earth. But the prevalence is relatively low [123,124]. It is possible that these animals had a different physiology compared to modern animals, including unique mechanisms for growth, metabolism, and reproduction. These animals may have had shorter lifespans that could have reduced the cumulative exposure to environmental carcinogens and the accumulation of cancer-associated mutations. They could have evolved adaptations to mitigate cancer risk, such as efficient DNA repair mechanisms or enhanced immune surveillance [125]. It is important to remember that the fossil record is incomplete and does not accurately represent the true prevalence of cancer in extinct animals due to preservational biases.

Soft tissue tumours are rarely preserved in fossils, and even when present, they may not be recognisable as cancer [126].

Numerous evolutionary hypotheses have addressed this question. Neoplastic disease may have first developed during the emergence of complex multicellular organisms and, more recently, evolved in organs that are more prone to oncogenesis [114]. Neoplastic growths may have some evolutionary value as they may develop during upbringing and introduce newly evolved genes to the gene pool and the development of the immune system [115].

Ancient records show examples of species that adopt some types of tumours as a biological strategy. For example, the fossil osteichthyan *Pachylebias* that existed in the hyper-saline water of the Mediterranean Sea 8 million years ago developed diffuse hyperostosis, which is a form of benign tumour originating from bone tissue (pachyostosis), to help smooth immersion and swimming in highly dense water by increasing the weight of their skeletons [116]. Synapsids of the Sirenidae group from 30 million years ago (Oligocene) also adopted a similar strategy to the Osteichthyes, to help them acquire high-density bone to browse at the bottom of shallow waters [63].

Overall, incorporating Peto’s Paradox into the study of extinct animals enriches our understanding of cancer biology and evolutionary adaptations across diverse taxa spanning millions of years of Earth’s history [127].

Comparing tumours found in fossils to those in modern organisms can shed light on the evolutionary history of diseases and the similarities and differences between ancient and contemporary biological processes. Studying tumours across different species of organisms can provide insights into their taxonomic relationships and evolutionary history. Similarities or differences in the types of tumours found in related species can indicate shared genetic traits or divergent evolutionary paths [128]. Osteosarcomas, for example, have been found in ancient and contemporary vertebrates [99].

When comparing the prevalence of tumours in fossils with currently existing animals, the most appropriate is to compare them with wild animals [128]. In modern wild animals, neoplasms are rare, sporadic in amphibians and sauropsids and slightly more frequent in osteichthyans and Synapsids. Cancer in wildlife still goes largely undetected [128]. Bone cancer has been reported in some wild animals, though the prevalence is not as well documented [129]. The epidemic is entirely different in domestic animals, such as dogs and cats, with tumours being more common and similar to the ones that occur in humans. This can be attributed to the fact that these animals now have longer life spans and live with humans exposed to the same environmental factors [122,130]. This increase in tumours is unprecedented and associated with human activity and new lifestyles. Osteosarcoma, described in several fossils that were likely connected with earlier peoples, is relatively common in dogs, particularly in larger breeds [131].

## 6. Conclusions

Comparative oncology explores cancer across various species, spanning living and extinct organisms. This interdisciplinary approach yields valuable insights into the commonalities and distinctions within cancer biology, enhancing our understanding of the disease’s fundamental nature and manifestations across the animal kingdom.

Although some studies have already explored this theme, our current study provides a thorough compilation, with revisited examples of cancer in paleontological samples that have come forth and some obscure examples that were lost in the grey literature. The data were organised chronologically to provide the reader with a view of tumours’ presence through time and not by vertebrate class.

The investigation of cancer in fossilised specimens faces significant hurdles. One of the primary challenges is the scarcity of well-preserved fossils, which limits the availability of specimens for study. Additionally, diagnosing diseases in ancient organisms is inherently difficult due to the degradation of biological material over time (millions of years) and the limited context for interpreting signs of disease.

From the 72 specimens included in this study, sauropsids were the most affected group, and the majority of the tumours described were malignant. Osteoma was the most common type of tumour described in all vertebrate groups.

Despite these challenges, technological advancements and methodological approaches are progressively overcoming them. Improved imaging techniques, such as high-resolution CT scans and advances in molecular biology, enable researchers to identify and analyse pathologies in the fossil record with increasing accuracy and detail. These technological advancements and interdisciplinary research efforts expand our capacity to detect and understand cancer and other diseases in ancient life forms.

Furthermore, the comparative analysis of the tumour prevalence in fossils, wild animals, and domestic animals underscores the multifaceted nature of cancer evolution. It highlights the impact of environmental changes, lifestyle factors, and human influence on the development and detection of cancer. Understanding these dynamics is crucial for advancing our knowledge of cancer biology, improving cancer detection and treatment in animals and humans and conserving wildlife health.

## Figures and Tables

**Figure 1 animals-14-01474-f001:**
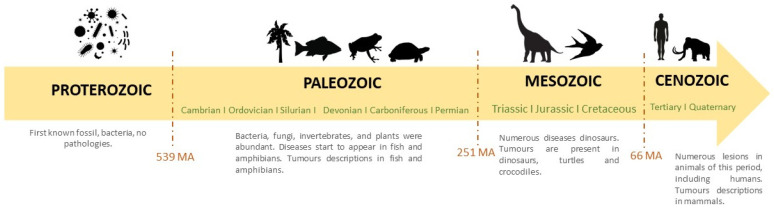
Paleontological timeline from the Proterozoic to the Cenozoic.

**Figure 2 animals-14-01474-f002:**
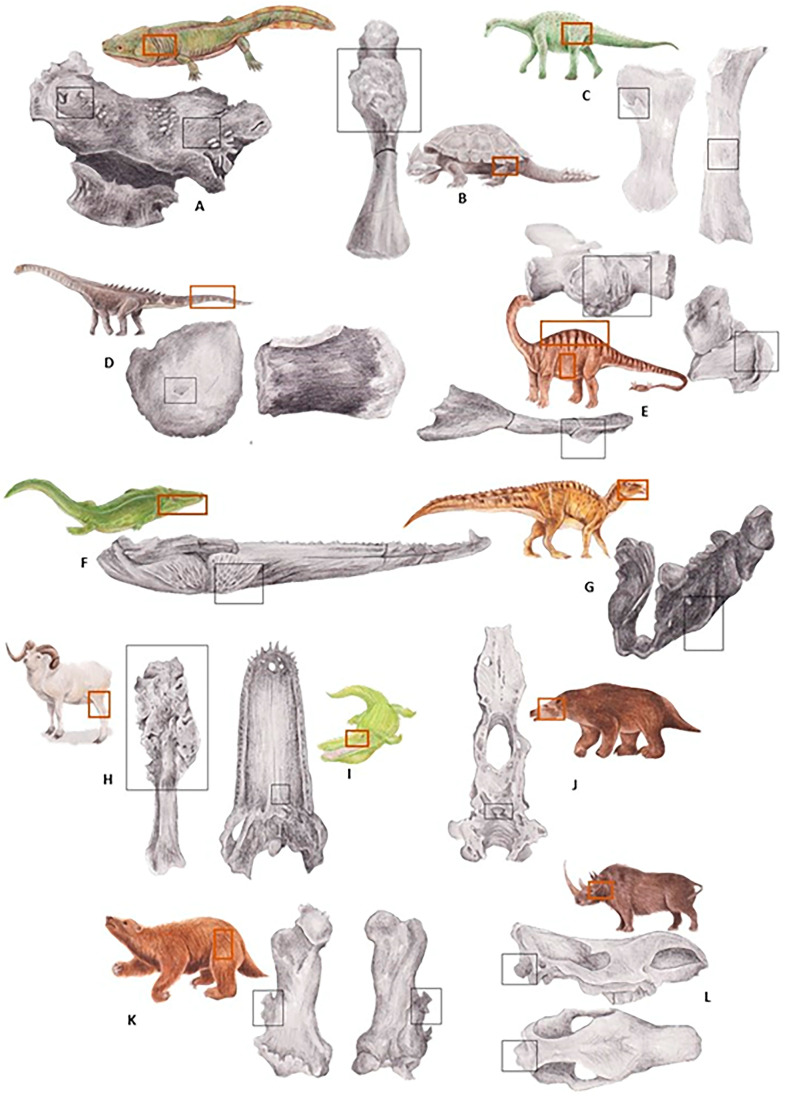
(**A**)—Osteosarcoma in the vertebral intercentrum in *Metoposaurus krasiejowensis* (specimen ZPAL Ab III/2467); (**B**)—femur osteosarcoma in *Pappochelys rosinae* (Staatliches Museum für Naturkunde, Stuttgart, Germany); (**C**)—femur osteoblastic tumour in *Bonitasaura salgadoi* (MPCA 460); (**D**)—vertebral osteoma in Titanosauridae (UFRJ-DG 508-R); (**E**)—vertebral hemangioma and rib osteochondroma in *Apatosaurus* (N/A); (**F**)—non-odontogenic osteoma in the right mandibular branch of a *Benthosuchus korobkovi* (GGM-0277-14/PV-00650); (**G**)—lower jaw ameloblastoma in *Telmatosaurus transsylvanicus* (LPB (FGGUB) R.1305); (**H**)—tibia osteosarcoma in *Ovis aries palustris* (N/A); (**I**)—jaw osteoma in *Mourasuchus pattersoni* (MCNC-PAL-110-72V); (**J**)—pituitary tumour in *Valgipes bucklandi* (MCT 4272-M); (**K**)—femur osteosarcoma in *Nothrotherium maquinense* (MCT4230-M); (**L**)—osteoma in the skull of *Coelodonta antiquitatis* (GMM KGU n747). Illustration by Andreia Garcês based on photos of the fossil and a description of the species. Boxes in black refer to the lesion location in the bone, boxes in orange are the bone anatomical location on the animal.

**Figure 3 animals-14-01474-f003:**
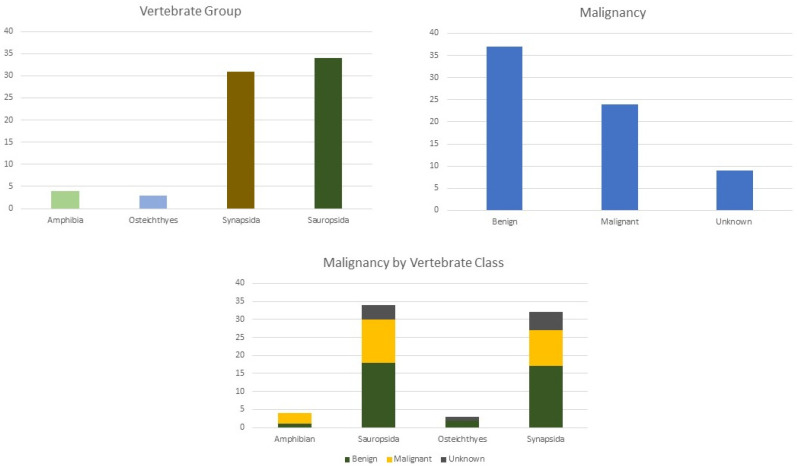
Distribution of tumour by vertebrate groups, malignancy, and malignancy by vertebrate class.

**Table 1 animals-14-01474-t001:** Summary of the main characteristics of different types of tumours described in fossils.

Tumour	Malignancy	Origin Cells	Affected Structures	Macroscopic Observation	Microscopic Observation	References
Osteoma	Benign	Osteoblasts	Mandibular bones, nasal sinuses, facial and cranial bones, limbs, sternum, ribs and skull	Delimited; covered with connective vascular tissue; at cut dense bone tissue, with fibrous connective tissue.	Osteoblasts and osteoclastic modelling form trabecular growth, with bone structures perpendicular to the surface of the tumour.	[30]
Osteosarcoma	Malignant	Mesenchymal stem cells	Long bones	Increased volume of the affected bone; congestion; oedema; presence of osteofibrous tissue; muscular atrophy; regional lymph nodes are enlarged and hard.	Depending on the dominance of a particular tissue, it can be classified as osteoblastic, chondroblastic or fibroblastic. Production of malignant osteoid cells with marked pleomorphism, varying in size, shape, and nuclear features. The tumour stroma may contain a mixture of spindle-shaped cells, multinucleated giant cells, and areas of necrosis.	[30]
Osteoblastic tumour	Benign	Osteoblasts	Spine, long bones	Well-defined, expansive masses within the bone; firm consistency.	Irregularly shaped trabeculae or sheets of woven bone interspersed with osteoblasts; cytologic atypia with enlarged nuclei and increased mitotic activity.	[31]
Odontoma	Benign	Odontoblasts, ameloblasts, and dental papilla cells.	Jaws, typically within the bone or embedded in the soft tissues surrounding developing teeth	Clusters or aggregates of denticles fused in a compact mass.	Multiple, small, tooth-like structures called denticles are organised arrangements of dental tissues (dentin, enamel, cementum, and pulp).	[32]
Ameloblastoma	Benign	Epithelial cells	Jaws	Single, well-defined masses or multicystic lesions with irregular borders. Vary in size and shape.	Islands or strands of epithelial cells arranged in a variety of patterns—follicular, plexiform, or acanthomatous arrangements, among others. A fibrous connective tissue stroma surrounds the epithelial islands.	[32]
Myeloma	Malignant	Plasma cells	Bone marrow	Bone destruction.	Presence of abnormal plasma cells.	[33]
Hemangioma	Benign	Endothelial cells	Vascular or cavernous neocapillaries	Aspect of a hemorrhagic mass, similar to telangiectatic osteosarcoma.	Necrotic foci and hemorrhagic spaces are found in the neoplastic mass, along with capillaries with irregular lumen, bordered by immature endothelial cells, in a fibrous stroma.	[34]

**Table 2 animals-14-01474-t002:** Neoplasia in fossils from Cambrian to Permian periods, by period, time (M.A.—millions of years), the country where it was found, species, vertebrate group (S—Synapsids; O—Osteichthyes and placoderms), anatomical region, type of tumour, and malignancy (B—benign, NA—unknown).

Period	Time (M.A.)	Country Where Found	Species	Vertebrate Group	Anatomical Localisation	Type of Tumour	Malignancy	Ref
Devonian	358–419	USA (Ohio)	*Dinichthys* spp.	O	Lower jawbone	Unknown	NA	[4,26]
Carboniferous	358–298	USA	*Phanerosteon mirabile*	O	Vertebrate	Osteoma	B	[53,54]
Permian	298–251	USA	*Gorgonopsian*	S	Canine root	Odontoma	B	[55]

**Table 3 animals-14-01474-t003:** Neoplasia in fossils from Triassic to Cretaceous periods, by period, time (M.A.—millions of years), the country where it was found, species, vertebrate group (A—amphibian, SA—sauropsids), anatomical region, type of tumour, and malignancy (B—benign, M—malign, NA—unknown).

Period	Time (M.A.)	Country Where Found	Species	Vertebrate Group	Anatomical Localisation	Type of Tumour	Malignancy	Ref
Triassic	225–215	Poland	*Metoposaurus krasiejowensis*	A	Vertebral intercentrum	Osteosarcoma	M	[59]
	225–215	Russia	*Parotosuchus* sp.	A	Craneal bone	Parostotic osteosarcoma	M	[60]
	225–215	Russia	*Benthosuchus korobkovi*	A	Right lower jaw	Non-odontogenic osteoma	B	[61]
	240	Germany	*Pappochelys rosinae*	SA	Femur	Osteosarcoma	M	[62]
Jurassic	157–146	USA	*Apatosaurus*	SA	Vertebrate	Hemangioma	B	[63]
	157–146	USA	*Apatosaurus*	SA	Rib	Osteochondroma	B	[26]
	161.5–145.0	China	*Gigantspinosaurus sichuanensis*	SA	Femur	Unknown	NA	[64]
	155–145	USA (Utah)	*Allosaurus fragilis*	SA	Humerus	Chondrosarcoma	M	[65]
	161–166	USA	Ceratopsia	SA	Skull	Myeloma	M	[66]
Cretaceous	145–55	Lebanon	Pycnodontiformes	A	The caudal tract of the vertebral column	Notochord chordoma	M	[67]
	145–66	Romania	*Telmatosaurus transsylvanica*	SA	Lower jaw	Ameloblastoma	B	[68]
	73–66	USA (Colorado)	*Edmontosaurus*	SA	Long bone	Metastatic cancer of sarcoma or osteosarcoma	M	[69]
	73–66	USA (Utah)	*Edmontosaurus*	SA	Vertebra	Hemangiosarcoma	M	[70]
	73–66	USA (Montana)	*Edmontosaurus*	SA	Vertebra	Osteoblastoma, hemangioma, desmoplastic	B	[15]
	73–66	China	*Lambeosaurinae bactrosaurus*	SA	Vertebra	Hemangioma	B	[15]
	81–76.7	Canada, USA	*Brachylophosaurus*	SA	Vertebra	Hemangioma	B	[15]
	81–76.7	Mongolia	*Gilmoreosaurus*	SA	Vertebra	Hemangioma	B	[15]
	73–66	USA	Mosasauridae	SA	Vertebra	Osteoma	B	[63]
	84–81	USA	*Platecarpus*	SA	Vertebra	Osteoma	B	[71]
	84–81	Patagonia (Argentina)	*Bonitasaura salgadoi*	SA	Femur	Osteoblastic tumour	B	[72]
	73–66	Argentina	*Bonapartesaurus rionegrensis*	SA	Metatarsal II	Unknown	NA	[13]
	73–66	Brazil	Titanosauridae	SA	Vertebra	Osteoma	B	[73]
	73–66	USA (Minnesota)	*Leidyosuchus* (*Borealosuchus*) *formidabilis*	SA	Ungual, phalanx, femora, scapula, vertebra	Osteoma	B	[74,75]
	76.5–75.5	Canada	*Centrosaurus apertus*	SA	Fibula	Osteosarcoma	M	[76]
	73–66	Canada	*Platercapus*	SA	Scapula	Osteoma	B	[26]
	83.6–72.1	Canada	*Stenonychosaurus inegualis*	SA	Cranial crest	Unknown	NA	[26]
	68–66	USA	*Triceratops*	SA	Scapula	Unknown	NA	[26]
	73–66	Canada (Alberta)	Hadrosauridae	SA	Vertebra	Langerhans Cell Histiocytosis (LCH)	M	[77]
	75	USA	*Centrosaurus apertus*	SA	Fibula	Osteosarcoma	M	[78]
	76.4–75.6	Canada	*Euoplocephalus*	SA	Vertebra	Unknown	B	[79]
	68–66	USA	*Torosaurus latus*	SA	Squamosal	Myeloma	M	[80]

**Table 4 animals-14-01474-t004:** Neoplasia in fossils from Cenozoic Era, by period, time (M.A.—millions of years), the country where it was found, species, vertebrate group (S—Synapsida, SA—Sauropsida, O—Osteichthyes and placoderms), anatomical region, type of tumour, and malignancy (B—benign, M—malign, NA—unknown).

Period	Time (M.A.)	Country Where Found	Species	Vertebrate Group	Anatomical Localisation	Type of Tumour	Malignancy	Ref
Eocene	56.0–33.9	Unknown	*Daphaenus* spp.	S	Unknown	Chondrosarcoma	M	[81]
	56.0–33.9	USA (Nebraska)	*Daphaenus* spp.	S	Jaw, teeth	Odontoma	B	[82,83]
Oligocene	33.9–23	Brazil	*Daphaenus felinus*	S	Radii	Unknown	NA	[71]
Miocene	23.03–5.333	USA	*Syllomus aegyptiacus*	SA	Shell	Osteoma	B	[84,85]
	23.03–5.333	Venezuela	*Mourasuchus pattersoni*	SA	Jaw	Osteoma/hamartoma	B	[86]
	23.03–5.333	USA	*Hesperocyon gregarius*	S	Radii	Osteochondroma	B	[87]
Pliocene	5.3–2.6	Chile	*‘Megaptera’ hubachi*	S	Skull	Osteoma	B	[88]
	5.33—2.6	Chile	*Balaenopteridae*	SA	Skull	Osteoma	B	[75]
Pleistocene	2.5–0.17	Brazil	*Valgipes bucklandi*	S	Basisphenoid	Pituitary tumour	N	[89]
	2.5–0.17	Russia, Poland	*Mammuthus primigenius*	S	Long limb bones, vertebrae, scapulae and ribs	Osteoid-osteoma, osteoblastoma	B	[90]
	2.5–0.17	Poland	*Mammuthus primigenius*	S	Ribs	Unknown	NA	[91]
	2.5–0.17	USA	*Mammuthus primigenius*	S	Tooth	Odontoma	B	[92]
	2.5–0.17	Poland	*Mammuthus primigenius*	S	Tooth	Osteoma	B	[93]
	2.5–0.17	Spain, France	Bovidae	S	Mandible	Osteoma, cyst	B	[94]
	2.5–0.17	South America	*Nothrotherium maquinense*	S	Jaw, teeth	Odontoma	B	[95]
	2.5–0.17	Brazil	*Nothrotherium maquinense*	S	Femur	Osteosarcoma	M	[96]
	2.5–0.17	France	*Ursus spelaeus*	S	Unknown	Benign tumour	B	[97]
	2.5–0.17	Slovenia	*Ursus spelaeus*	S	Jaw	Unknown	NA	[98]
	2.5–0.17	USA	*Ursus spelaeus*	S	Femur	Osteosarcoma	M	[99]
	2.5–0.17	Argentina	Ungulates	S	Tooth	Odontoma	B	[100]
	2.5–0.17	France	Equidae	S	Molar	Odontoma	B	[101]
	2.5–0.17	Japan	*Palaeoloxodon naumanni*	S	Tooth	Odontoma	B	[102]
	2.5–0.17	North America	*Bison latifrons*	S	Unknown	Osteosarcoma	M	[103]
	2.5–0.17	Russia	*Coelodonta antiquitatis*	S	Skull	Osteoma	B	[104]
	2.5–0.17	North America and Europe	*Aphanius crassicaudatus*	O	Vertebra	Osteoma	B	[26]

## Data Availability

Data are contained within the article.

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
