# Peer review of "Ancient Diseases in Vertebrates: Tumours through the Ages"

_animals, 2024, doi:10.3390/ani14101474_

Round 1
Reviewer 1 Report
Comments and Suggestions for Authors
The present manuscript describe the occurrences of tumors in animals throughout the time, from the Paleozoic to the present. The most relevant issue that I noticed is a repeatedly incongruence between the text and the references. Several statements present along the text are supported by papers that are not related to. Therefore, the authors should update the bibliography, adding papers that are congruent with the text and deleting unrelated ones (I have proposed some works that the authors should consider, but the text need more references).
The section "form Cambrian to Cretaceous..." is a bit confusing, because this chapter starts by mentioning occurrences found in more modern animals. The authors should move this entire section in the next chapter. Also the tables have several errors, regarding the geological time subdivisions and their ages.
There are also minor errors that I have marked in the pdf version, such as some species names are not in italic (species names always in italic) or species names are in uppercase (should be in lowercase), and portions of the text are repeated.
I look forward an improved version of the manuscript and I'm available if the authors need further information.

Author Response
The present manuscript describe the occurrences of tumors in animals throughout the time, from the Paleozoic to the present. The most relevant issue that I noticed is a repeatedly incongruence between the text and the references. Several statements present along the text are supported by papers that are not related to. Therefore, the authors should update the bibliography, adding papers that are congruent with the text and deleting unrelated ones (I have proposed some works that the authors should consider, but the text need more references).
Authors answer: the references were corrected
The section "form Cambrian to Cretaceous..." is a bit confusing, because this chapter starts by mentioning occurrences found in more modern animals. The authors should move this entire section in the next chapter. Also the tables have several errors, regarding the geological time subdivisions and their ages.
Authors answer: this section was improved, and divided in several sections.
There are also minor errors that I have marked in the pdf version, such as some species names are not in italic (species names always in italic) or species names are in uppercase (should be in lowercase), and portions of the text are repeated.
Authors answer: thank you for the comments the corrections were performed.
I look forward an improved version of the manuscript and I'm available if the authors need further information.
Reviewer 2 Report
Comments and Suggestions for Authors
Summary:
The article 'Ancient Diseases in Animals: tumors through the ages' is primarily a literature review of tumors and tumor-like pathologies in vertebrate animals from >>10,000 years ago. Two tables list most of these examples and the text largely interprets these findings or provides context for the study and what to make of it. The paper lists examples chronologically in most cases, rather than by taxonomy. No new evidence is presented, though some illustrations are new to the paper and highlight the appearance of some of the fossil cases of tumors.
Opinion:
The paper is satisfactory as a literature review. The novelty is not very high, as there are already several reviews of tumors in non-human animals and even a few reviews of tumors in the fossil record. Still, this could be a useful reference for researchers comparing their findings to the sum total of knowledge on this topic. The abstract and summary begin to explain that new advancements have improved our ability to make accurate diagnoses of fossil tumors, but I felt this was not clearly expanded upon in the main text. I propose some revisions to the paper below. Most are small changes, though I do propose a couple of areas to re-focus the writing and in one case propose an additional figure.
Strengths:
As a review, this document is quite thorough. There are >100 citations and this seems about right for the general nature of the review. I am most familiar with cases from reptiles and amphibians, so I note some possible missing records later in my comments. Otherwise, I was impressed with the lists.
The figures are quite beautiful and contribute to the character of the study. I can't be sure that the illustrations bring new knowledge, but they make a few of these obscure examples more accessible and easy to understand.
Weaknesses:
The cases are presented in chronological order, but it is not made clear why this is useful to do. I would expect a phylogenetic inference to be made from these observations, but listing them chronologically didn't much help with that. My suggestion here is that if the authors are committed to a timeline of observations as exists in Tables 1 and 2 and in the main text, supplement these observations with a new figure. This figure would show a typical paleontological timeline (in this case starting roughly in the Devonian) with four rows. In each row, insert a vertical line or box representing the timing of an observation for the four phylogenetic groups. This would more easily allow a reader to intuit the relative abundance of records across time for these groups compared to reading them as a table and mixed with all other taxa. Alternatively, it may be more interesting to a reader to view Tables 1 and 2 grouped by taxonomy rather than by time before present day. In either case, I recommend thinking hard about what pattern should be conveyed to the reader and how can we most easily communicate this information.
As previously noted, the abstract and summary indicate recent technological advances have improved our understanding of this topic. It is unclear from the tables, at least, how this assertion contributes to the current review. I propose that if the authors know that any of these cases are the result of a 'recent' investigation or re-analysis, they be indicated somehow with bold font, an asterisk, or similar marking.
Though it is hard to check every case in the timeline given for review, I think the publication adds very few new observations to this literature that already has many reviews on the topic. That's not necessarily a mark of a bad paper, but I didn't sense the authors did enough to explain what is new and interesting about this literature review. It would be enough to say something like 'existing reviews neglected a certain taxon' or 'since this review [citation], some new cases or revisited examples of cancer in paleontological samples have come forth.' For example, https://doi.org/10.1002/ar.24475 this publication produced a new example, but also in the review of literature it was noted that many examples cited are challenging references to find and acquire due to their age, obscurity, or being in languages not common to modern science. So bringing these examples to one place was a justification for publication.
Areas for Improvement:
The title of the paper suggests that non-vertebrate animals would also be included. They aren't. So I think the title should be revised to be more clear that this is 1) a review; and 2) focused on vertebrate animals.
In my observation, the authors do not appear to cite any of their own work. If they have any relevant work to cite, I think it is quite appropriate to do so. If not, that is obviously fine, too.
The authors repeat a sentiment in the literature that diagnosing osteosarcoma from fossil specimens is challenging. They also note in lines 256-259 that osteosarcoma is a common disease in modern canines. It is challenging to rectify these two assertions. I encourage the authors to read (https://doi.org/10.1002/ar.24475) as this includes references for many observations of bony tumors, including osteosarcoma, in reptiles. A few examples from the paleo record are also cited.
Mostly in Table 1, vertebrate group is listed as Fish, Amphibian, Reptile, or Mammalian. These are informal classifications and I think it would be more interesting to the biology-minded reader to instead classify as the larger phylogenetic group. For example, osteicthyes, sauropsida (as this includes birds), synapsida, amphibia (sensu lato). This would resolve any suggestion of misclassification across the 300 MY of paleontology that the authors summarize.
In line 366, the authors assert that advances in molecular biology my assist with the identification of pathology in the fossil record. I don't know how that could happen. If the authors feel that this is actually possible and relevant, I think it bears further comment.
The figures 1-4 are quite fabulous. I am not easily able to tie these illustrations back to the table and text, though. If the illustrations are based on cases listed in these sources, it would be helpful to cite the specimen either with a reference or specimen number from the museum or institution.
Related to the figures, the reconstructions are very beautiful, but I cannot tell how evidence is used to build them. A brief description of the illustrator's process for guiding these paleo reconstructions may help the reader understand their sources of information and potential biases. This could be at the end of the text, if it is awkward to insert this text in the main document.
Line Item changes:
1-2 - Use consistent title case for the title.
85 - replace 'consonant' with 'consistent'
101 - double check that this reference [27] is correct. I don't have a copy of it, but it is hard to see correspondence between this claim and the reference title.
118 - replace 'data' with 'date'
144 - replace 'Millian' with 'million'
148-149 - rephrase 'Still, they considered them all doubtful.' as 'Still, known examples are doubted.' Include a reference after this statement, if possible.
150 - replace 'inicial dignose' with 'the initial diagnosis'
151 - run-on sentence. Start the statement after 'and' as a new sentence.
152-154 - rephrase 'is insufficiently documented because [...]'. As-is, I am not sure of the meaning of this sentence nor why the authors claim that the case was insufficiently documented.
154-156 - The sentence beginning with 'In a diagnosis [...]', rephrase to make a sensible sentence. It is currently not written correctly and is therefore confusing.
168 - I think reword how the mammal orders "appeared," and instead state 'All present-day mammalian orders are represented in the Quaternary (2.58 million years ago) up to today.' Perhaps there is another way to say this. It's not really clear to me.
218-227 - This section references [91]. I checked this reference. I think the authors meant to also cite [90], as this is the paper with the technical description of pathology.
259-266 - Italics are used throughout this section, but inconsistently. Use italics only for species names.
277-278 and 301-302 - A nearly identical sentence is re-used. One could probably be removed with no change in meaning.
340 - I think italics should be used for the species name
References to consider (not necessary to cite all of these!):
Anné, J., Garwood, R. J., Lowe, T., Withers, P. J., & Manning, P. L. (2015). Interpreting pathologies in extant and extinct archosaurs using micro-CT. PeerJ, 3, e1130.
Faarke, A. A., & O'Connor, P. M. (2007). Pathology in Majungasaurus crenatissimus (Theropoda: Abelisauridae) from the late cretaceous of Madagascar. Journal of Vertebrate Paleontology, S2, 180–184
Frye, F. L. (1991b). Common pathological lesions and disease processes: Neoplasia. In F. L. Frye (Ed.), Reptile care: An atlas of diseases and treatments (Vol. 2, pp. 576–609). Neptune City, NJ: THF Publishing.
Frye, F. L. (1994). Diagnosis and surgical treatment of reptilian neoplasms with a compilation of cases 1966-1993. In Vivo, 8, 885–892.
Garner, M. M., Hernandez-Divers, S. M., & Raymond, J. T. (2004). Reptile neoplasia: A retrospective study of case submissions to a specialty diagnostic service. Veterinary Clinics of North America: Exotic Animal, 7, 653–671.
Hall, A. S., J L. Jacobs, & E. N. Smith. (2020). Possible osteosarcoma reported from a new world elapid snake and review of reptilian bony tumors. The Anatomical Record, 2020, 1-20.
Harshbarger, J. C. (1969). The registry of tumors in lower animals (Vol. 31, pp. 11–16). Bethesda, MD: National Cancer Institute Monograph.
Harshbarger, J. C. (1974). Activities report registry of tumors in lower animals 1965–1973. Washington, DC: Smithsonian Institution Press.
Harshbarger, J. C. (1975). Activities report registry of tumors in lower animals: 1974 supplement. Washington, DC: Smithsonian Institution Press.
Harshbarger, J. C. (1977). Role of the registry of tumors in loweranimals in the study of environmental carcinogenesis inaquatic animals. Annals of the New York Academy of Sciences, 298, 280–289.
Hernandez-Divers, S. M., & Garner, M. M. (2003). Neoplasia of reptiles with an emphasis on lizards. Veterinary Clinics of North America: Exotic Animal, 6, 251–273.
Machotka, S. V. (1984). Neoplasia in reptiles. In G. L. Hoff, F. L. Frye, & E. R. Jacobson (Eds.), Diseases of amphibians and reptiles. New York, NY: Plenum Press.
Mauldin, G. N., & Done, L. B. (2006). Oncology. In D. R. Mader (Ed.), Reptile medicine and surgery (2nd ed., pp. 299–322). Saunders Elsevier: St. Louis, MO.
Moodie, R. L. (1917). Studies in paleopathology. I. General consideration of the evidence of pathological conditions found among fossil animals. Annals of Medical History, 1917, 374–381.
Rothschild, B. M., Schultze, H.-P., & Pellegrini, R. (2012). Herpetological osteopathology: Annotated bibliography of amphibians and reptiles. New York, NY: Springer.
The paper is generally well written, but the middle section (parts 2 and 3) in particular contains several examples of poor grammar, incomplete sentences, typos, and other things like that. I listed a few specific items by line number, but it would be good to read through the manuscript very carefully before final submission as proofs to avoid mistakes like this.
I did not check the literature cited section for spelling, grammar, consistency, or style issues. Review this section carefully with the publisher before publication.
Author Response
Summary:
The article 'Ancient Diseases in Animals: tumors through the ages' is primarily a literature review of tumors and tumor-like pathologies in vertebrate animals from >>10,000 years ago. Two tables list most of these examples and the text largely interprets these findings or provides context for the study and what to make of it. The paper lists examples chronologically in most cases, rather than by taxonomy. No new evidence is presented, though some illustrations are new to the paper and highlight the appearance of some of the fossil cases of tumors.
Opinion:
The paper is satisfactory as a literature review. The novelty is not very high, as there are already several reviews of tumours in non-human animals and even a few reviews of tumors in the fossil record. Still, this could be a useful reference for researchers comparing their findings to the sum total of knowledge on this topic. The abstract and summary begin to explain that new advancements have improved our ability to make accurate diagnoses of fossil tumors, but I felt this was not clearly expanded upon in the main text. I propose some revisions to the paper below. Most are small changes, though I do propose a couple of areas to re-focus the writing and in one case propose an additional figure.
Strengths:
As a review, this document is quite thorough. There are >100 citations and this seems about right for the general nature of the review. I am most familiar with cases from reptiles and amphibians, so I note some possible missing records later in my comments. Otherwise, I was impressed with the lists.
The figures are quite beautiful and contribute to the character of the study. I can't be sure that the illustrations bring new knowledge, but they make a few of these obscure examples more accessible and easy to understand.
Author answer: thank you for the comment
Weaknesses:
The cases are presented in chronological order, but it is not made clear why this is useful to do. I would expect a phylogenetic inference to be made from these observations, but listing them chronologically didn't much help with that. My suggestion here is that if the authors are committed to a timeline of observations as exists in Tables 1 and 2 and in the main text, supplement these observations with a new figure. This figure would show a typical paleontological timeline (in this case starting roughly in the Devonian) with four rows. In each row, insert a vertical line or box representing the timing of an observation for the four phylogenetic groups. This would more easily allow a reader to intuit the relative abundance of records across time for these groups compared to reading them as a table and mixed with all other taxa. Alternatively, it may be more interesting to a reader to view Tables 1 and 2 grouped by taxonomy rather than by time before present day. In either case, I recommend thinking hard about what pattern should be conveyed to the reader and how can we most easily communicate this information.
Author answer: thank you for the suggestion in a initial manuscript we thought about describe by group of vertebrates but later chose by geological period we added the figure with timeline.
As previously noted, the abstract and summary indicate recent technological advances have improved our understanding of this topic. It is unclear from the tables, at least, how this assertion contributes to the current review. I propose that if the authors know that any of these cases are the result of a 'recent' investigation or re-analysis, they be indicated somehow with bold font, an asterisk, or similar marking.
Author answer: thank you for the suggestion it is little hard, since the information is not very clear in most article. A Section regarding diagnose was added as requested by reviewer 1 to help understand the process.
Though it is hard to check every case in the timeline given for review, I think the publication adds very few new observations to this literature that already has many reviews on the topic. That's not necessarily a mark of a bad paper, but I didn't sense the authors did enough to explain what is new and interesting about this literature review. It would be enough to say something like 'existing reviews neglected a certain taxon' or 'since this review [citation], some new cases or revisited examples of cancer in paleontological samples have come forth.' For example, https://doi.org/10.1002/ar.24475 this publication produced a new example, but also in the review of literature it was noted that many examples cited are challenging references to find and acquire due to their age, obscurity, or being in languages not common to modern science. So bringing these examples to one place was a justification for publication.
Areas for Improvement:
The title of the paper suggests that non-vertebrate animals would also be included. They aren't. So I think the title should be revised to be more clear that this is 1) a review; and 2) focused on vertebrate animals.
In my observation, the authors do not appear to cite any of their own work. If they have any relevant work to cite, I think it is quite appropriate to do so. If not, that is obviously fine, too.
Author answer: the title was change. Unfortunately the authors do not have any paper in the area, only in living animal that do not make sense include in this apper.
The authors repeat a sentiment in the literature that diagnosing osteosarcoma from fossil specimens is challenging. They also note in lines 256-259 that osteosarcoma is a common disease in modern canines. It is challenging to rectify these two assertions. I encourage the authors to read (https://doi.org/10.1002/ar.24475) as this includes references for many observations of bony tumors, including osteosarcoma, in reptiles. A few examples from the paleo record are also cited.
Author answer: That section was eliminated
Mostly in Table 1, vertebrate group is listed as Fish, Amphibian, Reptile, or Mammalian. These are informal classifications and I think it would be more interesting to the biology-minded reader to instead classify as the larger phylogenetic group. For example, osteicthyes, sauropsida (as this includes birds), synapsida, amphibia (sensu lato). This would resolve any suggestion of misclassification across the 300 MY of paleontology that the authors summarize.
Authors answer: thank you for the comments the new phylogenetic groups were added.
In line 366, the authors assert that advances in molecular biology my assist with the identification of pathology in the fossil record. I don't know how that could happen. If the authors feel that this is actually possible and relevant, I think it bears further comment.
Author answer: a section regarding technologies was added.
The figures 1-4 are quite fabulous. I am not easily able to tie these illustrations back to the table and text, though. If the illustrations are based on cases listed in these sources, it would be helpful to cite the specimen either with a reference or specimen number from the museum or institution.
Related to the figures, the reconstructions are very beautiful, but I cannot tell how evidence is used to build them. A brief description of the illustrator's process for guiding these paleo reconstructions may help the reader understand their sources of information and potential biases. This could be at the end of the text, if it is awkward to insert this text in the main document.
Authors answer: thank you for the suggestion the information was added to the text.
Line Item changes:
1-2 - Use consistent title case for the title.
85 - replace 'consonant' with 'consistent'
101 - double check that this reference [27] is correct. I don't have a copy of it, but it is hard to see correspondence between this claim and the reference title.
118 - replace 'data' with 'date'
144 - replace 'Millian' with 'million'
148-149 - rephrase 'Still, they considered them all doubtful.' as 'Still, known examples are doubted.' Include a reference after this statement, if possible.
150 - replace 'inicial dignose' with 'the initial diagnosis'
151 - run-on sentence. Start the statement after 'and' as a new sentence.
152-154 - rephrase 'is insufficiently documented because [...]'. As-is, I am not sure of the meaning of this sentence nor why the authors claim that the case was insufficiently documented.
154-156 - The sentence beginning with 'In a diagnosis [...]', rephrase to make a sensible sentence. It is currently not written correctly and is therefore confusing.
168 - I think reword how the mammal orders "appeared," and instead state 'All present-day mammalian orders are represented in the Quaternary (2.58 million years ago) up to today.' Perhaps there is another way to say this. It's not really clear to me.
218-227 - This section references [91]. I checked this reference. I think the authors meant to also cite [90], as this is the paper with the technical description of pathology.
259-266 - Italics are used throughout this section, but inconsistently. Use italics only for species names.
277-278 and 301-302 - A nearly identical sentence is re-used. One could probably be removed with no change in meaning.
340 - I think italics should be used for the species name
References to consider (not necessary to cite all of these!):
Anné, J., Garwood, R. J., Lowe, T., Withers, P. J., & Manning, P. L. (2015). Interpreting pathologies in extant and extinct archosaurs using micro-CT. PeerJ, 3, e1130.
Faarke, A. A., & O'Connor, P. M. (2007). Pathology in Majungasaurus crenatissimus (Theropoda: Abelisauridae) from the late cretaceous of Madagascar. Journal of Vertebrate Paleontology, S2, 180–184
Frye, F. L. (1991b). Common pathological lesions and disease processes: Neoplasia. In F. L. Frye (Ed.), Reptile care: An atlas of diseases and treatments (Vol. 2, pp. 576–609). Neptune City, NJ: THF Publishing.
Frye, F. L. (1994). Diagnosis and surgical treatment of reptilian neoplasms with a compilation of cases 1966-1993. In Vivo, 8, 885–892.
Garner, M. M., Hernandez-Divers, S. M., & Raymond, J. T. (2004). Reptile neoplasia: A retrospective study of case submissions to a specialty diagnostic service. Veterinary Clinics of North America: Exotic Animal, 7, 653–671.
Hall, A. S., J L. Jacobs, & E. N. Smith. (2020). Possible osteosarcoma reported from a new world elapid snake and review of reptilian bony tumors. The Anatomical Record, 2020, 1-20.
Harshbarger, J. C. (1969). The registry of tumors in lower animals (Vol. 31, pp. 11–16). Bethesda, MD: National Cancer Institute Monograph.
Harshbarger, J. C. (1974). Activities report registry of tumors in lower animals 1965–1973. Washington, DC: Smithsonian Institution Press.
Harshbarger, J. C. (1975). Activities report registry of tumors in lower animals: 1974 supplement. Washington, DC: Smithsonian Institution Press.
Harshbarger, J. C. (1977). Role of the registry of tumors in loweranimals in the study of environmental carcinogenesis inaquatic animals. Annals of the New York Academy of Sciences, 298, 280–289.
Hernandez-Divers, S. M., & Garner, M. M. (2003). Neoplasia of reptiles with an emphasis on lizards. Veterinary Clinics of North America: Exotic Animal, 6, 251–273.
Machotka, S. V. (1984). Neoplasia in reptiles. In G. L. Hoff, F. L. Frye, & E. R. Jacobson (Eds.), Diseases of amphibians and reptiles. New York, NY: Plenum Press.
Mauldin, G. N., & Done, L. B. (2006). Oncology. In D. R. Mader (Ed.), Reptile medicine and surgery (2nd ed., pp. 299–322). Saunders Elsevier: St. Louis, MO.
Moodie, R. L. (1917). Studies in paleopathology. I. General consideration of the evidence of pathological conditions found among fossil animals. Annals of Medical History, 1917, 374–381.
Rothschild, B. M., Schultze, H.-P., & Pellegrini, R. (2012). Herpetological osteopathology: Annotated bibliography of amphibians and reptiles. New York, NY: Springer.
Author answer: thank you for the suggestion all corrections were added.
Reviewer 3 Report
Comments and Suggestions for Authors
This review presents a summary of findings in the field of “Paleo-oncology”, the study of neoplastic lesions found in fossil and/or ancient organisms. The topic is interesting and as such there might be some value in publishing this paper. However, the paper, in its present form is not acceptable for publications for the following reasons:
1) The paper is poorly organized and quite enumerative, in particular note that lines 269-292 are copied identically I n lines 293-316, however, the reference cited are different !!!! This casts a serious question on the list of references presented, the authors should at least re-read a paper before sending it out, I did not check each reference, but it should be done to avoid errors. Most of the example cited in more detail refer to recent species (few thousands years-old specimens), some examples of really fossil material could help.
2) Some critical informations are missing, for example in the Conclusions there is reference to technological and methodological advances which have permitted to improve our understanding of paleo-oncological lesions. These techniques should be briefly described in detail at the beginning and examples of their use should be given, possibly with figures. What about genetics? An example is given for Canine Transmisible Venereal Tumor, but is this paleo-oncology? If yes one should explain better how the study of the evolution of specific variants of a gene is studied and how this knowledge can lead to the study of present lesions(not only for CTVT, but also for other tumors). Although genetics has in general not much to do with paleontology many interesting example exists, but some discussion is needed.
3) Figure 1-4 have NO REFERENCES, and look like the pictures shown in books of my grandson…As such they have no interest. One should show good specific examples with high-resolution CT scans, possibly sections or any other type of analysis. It is essential to provide diagnostic clues which permit to identify different types of lesions.
4) As the paper is for non-specialists, one should explain which are the different tumors found: what is the difference between an osteoma and an osteosarcoma, what does it mean to find these different types of tumors. How are they diagnosed? What about chondrosarcoma, how can this tumor be found in fossils? And other tumors such as pituitary tumors?
5) No attempt is done to make some statistics or meta-analysis it is only repeated that these tumors are rare and a reference is given to the Pet`s paradox (that is actually valid for existent organisms as well) and to the differdence between domesticated and non-domesticated species. How can really paleo-oncology help in understanding environmental effects? It is not clear.
6) The paper is poorly written and the sentences are sometimes difficult to understand.
This paper should not be published in its present form, but could be reconsidered after major revisions.
Comments on the Quality of English LanguageThe paper is poorly written and must be revised by a native speaker. Many sentences are really difficult to understand and/or incorrect
Author Response
This review presents a summary of findings in the field of “Paleo-oncology”, the study of neoplastic lesions found in fossil and/or ancient organisms. The topic is interesting and as such there might be some value in publishing this paper. However, the paper, in its present form is not acceptable for publication for the following reasons:
- The paper is poorly organized and quite enumerative, in particular note that lines 269-292 are copied identically in lines 293-316, however, the references cited are different !!!! This casts a serious question on the list of references presented, the authors should at least re-read a paper before sending it out, I did not check each reference, but it should be done to avoid errors. Most of the examples cited in more detail refer to recent species (a few thousand years-old specimens), some examples of real fossil material could help).
Author answer: thank you for the comments, we apologize for the error in the the references. We corrected all the references again. The text was improved in many sections.
- Some critical information is missing, for example in the Conclusions there is a reference to technological and methodological advances which have permitted us to improve our understanding of paleo-oncological lesions. These techniques should be briefly described in detail at the beginning and examples of their use should be given, possibly with figures. What about genetics? An example is given for Canine Transmissible Venereal Tumor, but is this paleo-oncology? If yes one should explain better how the study of the evolution of specific variants of a gene is studied and how this knowledge can lead to the study of present lesions(not only for CTVT but also for other tumours). Although genetics has in general not much to do with paleontology many interesting example exists, but some discussion is needed.
Author answer: thank you for your suggestion. Technological and methodological advances were added after the introduction of a new topic. Also, more information regarding the importance of genetics in tumour detection has been added.
3) Figures 1-4 have NO REFERENCES, and look like the pictures shown in books of my grandson…As such they have no interest. One should show good specific examples with high-resolution CT scans, possibly sections or any other type of analysis. It is essential to provide diagnostic clues which permit to identification of different types of lesions.
Author answers: the figures are from my authorship based on photographs and descriptions of the articles. Specific examples with high-resolution CT scans would be complicated to add since many do not have that data available. Also, the authors think that the illustration adds more information to the manuscript and makes it less monotonous. The images were improved by request of reviewer 2.
- As the paper is for non-specialists, one should explain which are the different tumors found: what is the difference between an osteoma and an osteosarcoma, what does it mean to find these different types of tumors. How are they diagnosed? What about chondrosarcoma, how can this tumor be found in fossils? And other tumors such as pituitary tumors?
Author answers: a new section was added with this information.
5) No attempt is done to make some statistics or meta-analysis it is only repeated that these tumors are rare and a reference is given to the Pet`s paradox (that is actually valid for existent organisms as well) and to the differdence between domesticated and non-domesticated species. How can really paleo-oncology help in understanding environmental effects? It is not clear.
Author answers: a new section was added with this information with statistic on the paper added and the discussion was improved to address this questions.
6) The paper is poorly written and the sentences are sometimes difficult to understand.
This paper should not be published in its present form, but could be reconsidered after major revisions.
Author answers: the English was corrected.
Reviewer 4 Report
Comments and Suggestions for Authors
The authors have put forth a good concept with collecting information on tumours in animals through time, and they have collected a fair amount of good information regarding this concept. However, there are a number of things with the study/paper that need to be fixed or improved so that this can be conveyed to the reader in the clearest way possible.
The various sections should have a consistency to them. In saying that, the Cenozoic gets its own section, but the two earlier eras of the Phanerozoic Eon are piled together. While I understand that is partially because more is known about the Cenozoic, these sections should still be consistent with each era getting its own section. Separate out the Paleozoic and Mesozoic from each other. This may require a bit more to be stated about the occurrences/records within those eras to ensure each section is large enough, but should help make the paper more complete. This should also result in Table 1 being split into two tables.
There are a number of instances where subfossil and/or archaeological records are mentioned in the text. These are listed in no table and it is not clear how they coincide with the other, older records. While these would still be considered part of the Cenozoic, I suppose the reason for not listing them is because they would be far more numerous. The Cenozoic may be more about everything prior to the Holocene then, but this should all be clarified. Even without a table, it may be better to have a mainly Cenozoic section, and then a Holocene to Recent section.
There is a lack of consistency in the terms that are used. For instance, the term tumor and tumour are both used throughout the paper at various places. I have marked some of these to highlight them, but one should be chosen and used to the exclusion of the other throughout. The same can be said of other terms like etiology versus aetiology, or the use of the additional "a" in words like palaeontology versus paleontology and Paleozoic versus Palaeozoic, etc.
There are two instances in the Discussion where two paragraphs are repeated back to back.
There are a number of instances in the Cancer through Time section, where previous papers and authors are referred to with no in-text citations or references provided. These must be added and clarified in the text and perhaps to the References section.
The References section has a clear lack of consistency in the formatting of the various sources. I marked a few of these in the annotated PDF, but this whole section must be reviewed to ensure it follows the same formatting throughout.
The figures are small and don't do a good job at utilizing the space available. The figures should be rearranged, and potentially some should be combined (particularly Figs. 3 and 4) to remove significant blank space and maximize the space available. Another option would be to enlarge the figures, which would also help the reader see things more clearly. While specimen numbers for the figured specimens should be added, there should also be scale bars to provide an idea as to the size of the specimens. Additionally, it appears that the reconstructions of the various taxa are taken from online rather than drawn by the authors. If that is the case, then the original authors must be credited and acknowledged for their work as well.
The authors could also include something more in the discussion about the prevalence of particular types of tumours (and potentially cancers). While this is briefly mentioned, expanding on that a bit more would help take this study from less of a review and more of an opportunity to provide something more novel with the study.
While there are a number of things to be done, I do think it would be useful to have this published, although these fixes need to be done to allow readers to get more from the study than they currently could.
I have included an annotated PDF with a number of comments and corrections on it as well. These include some of the things mentioned here and hopefully will provide more direction with where to make changes and what needs to be done to help get this study into a better state for potential publication.

There seems to be random changing in the manuscript between some British English spellings and American English spellings. This needs to be fixed and made consistent.
Author Response
The authors have put forth a good concept with collecting information on tumours in animals through time, and they have collected a fair amount of good information regarding this concept. However, there are a number of things with the study/paper that need to be fixed or improved so that this can be conveyed to the reader in the clearest way possible.
Author answer: thank you for comments.
The various sections should have a consistency to them. In saying that, the Cenozoic gets its own section, but the two earlier eras of the Phanerozoic Eon are piled together. While I understand that is partially because more is known about the Cenozoic, these sections should still be consistent with each era getting its own section. Separate out the Paleozoic and Mesozoic from each other. This may require a bit more to be stated about the occurrences/records within those eras to ensure each section is large enough, but should help make the paper more complete. This should also result in Table 1 being split into two tables. There are a number of instances where subfossil and/or archaeological records are mentioned in the text. These are listed in no table and it is not clear how they coincide with the other, older records. While these would still be considered part of the Cenozoic, I suppose the reason for not listing them is because they would be far more numerous. The Cenozoic may be more about everything prior to the Holocene then, but this should all be clarified. Even without a table, it may be better to have a mainly Cenozoic section, and then a Holocene to Recent section.
Author answer: thank you fro the comments new sections were added to the text.
There is a lack of consistency in the terms that are used. For instance, the term tumor and tumour are both used throughout the paper at various places. I have marked some of these to highlight them, but one should be chosen and used to the exclusion of the other throughout. The same can be said of other terms like etiology versus aetiology, or the use of the additional "a" in words like palaeontology versus paleontology and Paleozoic versus Palaeozoic, etc.
Author answer: the manuscript was corrected
There are two instances in the Discussion where two paragraphs are repeated back to back.
Author answer: the discussion was improved has requested by Reviwer.
There are a number of instances in the Cancer through Time section, where previous papers and authors are referred to with no in-text citations or references provided. These must be added and clarified in the text and perhaps to the References section.
Author answer: the manuscript was corrected
The References section has a clear lack of consistency in the formatting of the various sources. I marked a few of these in the annotated PDF, but this whole section must be reviewed to ensure it follows the same formatting throughout.
Author answer: the manuscript was corrected
The figures are small and don't do a good job at utilizing the space available. The figures should be rearranged, and potentially some should be combined (particularly Figs. 3 and 4) to remove significant blank space and maximize the space available. Another option would be to enlarge the figures, which would also help the reader see things more clearly. While specimen numbers for the figured specimens should be added, there should also be scale bars to provide an idea as to the size of the specimens. Additionally, it appears that the reconstructions of the various taxa are taken from online rather than drawn by the authors. If that is the case, then the original authors must be credited and acknowledged for their work as well.
Author answer: Figures were collected together, regarding scale bars with was difficult. The images are from the author and based in information and paper an online, thecredits were added.
The authors could also include something more in the discussion about the prevalence of particular types of tumours (and potentially cancers). While this is briefly mentioned, expanding on that a bit more would help take this study from less of a review and more of an opportunity to provide something more novel with the study.
While there are a number of things to be done, I do think it would be useful to have this published, although these fixes need to be done to allow readers to get more from the study than they currently could.
I have included an annotated PDF with a number of comments and corrections on it as well. These include some of the things mentioned here and hopefully will provide more direction with where to make changes and what needs to be done to help get this study into a better state for potential publication.
Author answer: Thank you for the comments all the corrections were incorporated.
Round 2
Reviewer 1 Report
Comments and Suggestions for Authors
Dear Editor and Authors,
I'm glad to see the changes made to the manuscript and the added sections as well. I have attached a revised version of the manuscript with few suggestions.
I consider the manuscript acceptable after minor revision.
I'm available if the authors needs more explanations.
Best regards,
Mattia

Author Response
Dear Editor and Authors,
I'm glad to see the changes made to the manuscript and the added sections as well. I have attached a revised version of the manuscript with few suggestions.
I consider the manuscript acceptable after minor revision.
I'm available if the authors needs more explanations.
Best regards,
Mattia
Author answer: thank you for all the comments and suggestions
Reviewer 2 Report
Comments and Suggestions for Authors
Summary of changes:
The authors revised several sections throughout the manuscript. Most, but not all, changes are highlighted. This includes new sections for a reorganized paper: Types of tumours in fossils, Technological and methodological advances in the detection of neoplasia in fossils, and Tumours Descriptions In Fossil Remains. The new sections include additional tables or reorganized tables. A new statistical analysis is also presented with a figure as is a timeline figure. About 10 references were added. With this new framing, the authors have made the effort to reorganize their observations. Observations are still generally in chronological order with an overlay of broad taxonomy.
Areas for Improvement:
In lines 114-120, 141-170, and later in 436-441, despite some references provided, I am highly skeptical that aDNA or proteomics are relevant to the fossil record for the purposes of identifying cancer. Both methods are highly limited in the timeline of vertebrate evolution to maybe the last 1 million years in exceptional cases. This isn't even commented on by the authors. Perhaps they feel this is obvious to the reader? But I think it is of lesser relevance than the other methods suggested and should be given this caveat. With the specific example of the woolly mammoth, the authors point out mitochondrial genomes have been sequenced from these extinct animals, but this leaves a gap in reasoning between the relevance of the mitochondrial genome and cancer. Mitochondria are important for most cell functioning, but they are mostly separate from the nuclear genome and so this feels like a large logical jump to me. In a second example of aDNA relevance, the authors cite a case of CTVT in canids. Here, population genetic evidence is presented. While I love population genetics as a field of research, I feel it is not really supported here with this framing as a tool to help study cancer in the fossil record.
In lines 136-140, I wonder if this one reference is sufficient to back up the argument that three different technologies (CT, MRI, XRF) are being advanced for the purpose of identifying cancer in fossil specimens. The authors could cite literature of 'neontology' with dry or fluid-preserved museum specimens, as I am familiar with this literature and know there are at least a few dozen references.
The paleo timelines presented in section 4 (starting line 184, ending line 235) feels hastily written and contains factual errors with secondary source and popular media citations (e.g., 48). These errors should exclude the manuscript from publication if not corrected. It feels like the authors used a search engine to back up these points rather than from a good citation on the matter. This appears to have led to confusion, such as including Placoderms and Sharks within the Osteicthyes. If the authors are not sure about these facts, they should seek an outsider to fact check them.
More generally, I think the term osteichthyes is not correctly used throughout the new manuscript and is a direct replacement for where the term 'fish' was used. These are not interchangeable. That includes in the new tables and figures. These must be checked before being accepted for publication, else the authors will unintentionally make false claims that are hard to correct.
The observations are still presented in chronological order and I still feel that it is not explained to the reader why this ordering is useful.
There are two Table 1's and two Table 2's.
Reference formatting is quite variable. Sometimes journals are abbreviated and other times not. Some references are in all-capital letters. Some references not in English are translated and others are not. DOI's or ISBN's are given for about 2/3 of references, but not all. This is very chaotic.
Unfortunately, the grammar and spelling seem worse in the revised version. It is challenging to point out every thing that can be improved, but I list some things below.
Line Item changes:
Table 1: for osteoma row and macroscopic observation column, the description is confusing with this punctuation or wording. Specifically "at cut dense bone tissue, with fibrous connective tissue"
Table 1: I think "None marrow" is a typo for "Bone marrow," but I am unsure
Table 1: "plasm cells" may be a typo
Tables 2-4: many typos in the table captions
Table 2 (~line 199) malignancy not listed for last example.
Figure 1: the new figure 1 contains several typos within the figure. E.g., "desises," "bacteria's," "descriptions" instead of "described"
46: Replace 'depending on' with 'by'
86: remove '79' in the text.
98-102: I think the references cited in this section may not be correct. They don't seem relevant?
114: Replace "In the years" with "Through the years"
121: Use "Histology" instead of "Histological"
180-182: broken grammar in this sentence
191 and 192: Replace "Osteicthyeses" with "Osteicthyes"
192: Replace 'ostriches' with 'osteicthyes'
195: unclear in this context what 'dominated' means. Were they more abundant than something else? Only among vertebrates?
196: replace "synapsida synapsid ancestors" with "synapsids"
204-212: replace 'sauropsidas' with 'sauropsids'
309: "malignant" misspelled
310-311: incomplete sentence or bad grammar. I am confused.
333: do we really mean "population?" I think a better word is needed.
426: I don't know this phrase 'grey bibliography.' Perhaps you mean 'grey literature' and/or undercited literature
Unfortunately, the grammar and spelling seem worse in the revised version. It is challenging to point out every thing that can be improved, but I listed some things as line item changes.
Author Response
Summary of changes:
The authors revised several sections throughout the manuscript. Most, but not all, changes are highlighted. This includes new sections for a reorganized paper: Types of tumours in fossils, Technological and methodological advances in the detection of neoplasia in fossils, and Tumours Descriptions In Fossil Remains. The new sections include additional tables or reorganized tables. A new statistical analysis is also presented with a figure as is a timeline figure. About 10 references were added. With this new framing, the authors have made the effort to reorganize their observations. Observations are still generally in chronological order with an overlay of broad taxonomy.
Areas for Improvement:
In lines 114-120, 141-170, and later in 436-441, despite some references provided, I am highly skeptical that aDNA or proteomics are relevant to the fossil record for the purposes of identifying cancer. Both methods are highly limited in the timeline of vertebrate evolution to maybe the last 1 million years in exceptional cases. This isn't even commented on by the authors. Perhaps they feel this is obvious to the reader? But I think it is of lesser relevance than the other methods suggested and should be given this caveat. With the specific example of the woolly mammoth, the authors point out mitochondrial genomes have been sequenced from these extinct animals, but this leaves a gap in reasoning between the relevance of the mitochondrial genome and cancer. Mitochondria are important for most cell functioning, but they are mostly separate from the nuclear genome and so this feels like a large logical jump to me. In a second example of aDNA relevance, the authors cite a case of CTVT in canids. Here, population genetic evidence is presented. While I love population genetics as a field of research, I feel it is not really supported here with this framing as a tool to help study cancer in the fossil record.
Author answer: thank you for the comment, the authors changed a little the text to show that in the future can be an important tool.
In lines 136-140, I wonder if this one reference is sufficient to back up the argument that three different technologies (CT, MRI, XRF) are being advanced for the purpose of identifying cancer in fossil specimens. The authors could cite literature of 'neontology' with dry or fluid-preserved museum specimens, as I am familiar with this literature and know there are at least a few dozen references.
Author answer: the information was added.
The paleo timelines presented in section 4 (starting line 184, ending line 235) feels hastily written and contains factual errors with secondary source and popular media citations (e.g., 48). These errors should exclude the manuscript from publication if not corrected. It feels like the authors used a search engine to back up these points rather than from a good citation on the matter. This appears to have led to confusion, such as including Placoderms and Sharks within the Osteicthyes. If the authors are not sure about these facts, they should seek an outsider to fact check them.
Author answer: the information was checked.
More generally, I think the term osteichthyes is not correctly used throughout the new manuscript and is a direct replacement for where the term 'fish' was used. These are not interchangeable. That includes in the new tables and figures. These must be checked before being accepted for publication, else the authors will unintentionally make false claims that are hard to correct.
Authors answer: osteichthyes is correct in part to this group was added Placoderms, since there is one specie include in the text that belong to this groups and the other are bony fish.
The observations are still presented in chronological order and I still feel that it is not explained to the reader why this ordering is useful.
Authors answer: information added.
There are two Table 1's and two Table 2's.
Authors answer: corrected
Reference formatting is quite variable. Sometimes journals are abbreviated and other times not. Some references are in all-capital letters. Some references not in English are translated and others are not. DOI's or ISBN's are given for about 2/3 of references, but not all. This is very chaotic.
Authors answer: corrected
Unfortunately, the grammar and spelling seem worse in the revised version. It is challenging to point out everything that can be improved, but I list some things below.
Authors answer: corrected
Line Item changes:
Table 1: for osteoma row and macroscopic observation column, the description is confusing with this punctuation or wording. Specifically "at cut dense bone tissue, with fibrous connective tissue"
Table 1: I think "None marrow" is a typo for "Bone marrow," but I am unsure
Table 1: "plasm cells" may be a typo
Tables 2-4: many typos in the table captions
Table 2 (~line 199) malignancy not listed for last example.
Figure 1: the new figure 1 contains several typos within the figure. E.g., "desises," "bacteria's," "descriptions" instead of "described"
46: Replace 'depending on' with 'by'
86: remove '79' in the text.
98-102: I think the references cited in this section may not be correct. They don't seem relevant?
114: Replace "In the years" with "Through the years"
121: Use "Histology" instead of "Histological"
180-182: broken grammar in this sentence
191 and 192: Replace "Osteicthyeses" with "Osteicthyes"
192: Replace 'ostriches' with 'osteicthyes'
195: unclear in this context what 'dominated' means. Were they more abundant than something else? Only among vertebrates?
196: replace "synapsida synapsid ancestors" with "synapsids"
204-212: replace 'sauropsidas' with 'sauropsids'
309: "malignant" misspelled
310-311: incomplete sentence or bad grammar. I am confused.
333: do we really mean "population?" I think a better word is needed.
426: I don't know this phrase 'grey bibliography.' Perhaps you mean 'grey literature' and/or undercited literature
Authors answer: corrected
Reviewer 3 Report
Comments and Suggestions for Authors
The paper is now fully revised and can be accepted for publication.
Comments on the Quality of English LanguageGood quality of english.
Author Response
Thank you
Reviewer 4 Report
Comments and Suggestions for Authors
I commend the authors as this is a significant improvement on the previous version of the manuscript. There are a number of good additions, and these have helped make the paper far more complete. It should, hopefully, be fairly clear that I don't have anything against what is being written. There are a couple places where the grammar needs some help, or some small issues with consistency. I have attempted to go through the manuscript and make note of these in the annotated PDF that is attached. I think that if these things get changed/fixed by the authors, this paper will be ready for publication and will be a nice addition to the scientific literature on the subjects of oncology and paleo-oncology in particular.

As noted above, there are just a few grammatical issues and consistency issues, but I have attempted to make note of these in the annotated PDF.
Author Response
I commend the authors as this is a significant improvement on the previous version of the manuscript. There are a number of good additions, and these have helped make the paper far more complete. It should, hopefully, be fairly clear that I don't have anything against what is being written. There are a couple places where the grammar needs some help, or some small issues with consistency. I have attempted to go through the manuscript and make note of these in the annotated PDF that is attached. I think that if these things get changed/fixed by the authors, this paper will be ready for publication and will be a nice addition to the scientific literature on the subjects of oncology and paleo-oncology in particular.
Authors answer: corrected